

# An ensemble probabilisitic approach to reconstruct the biogeochemical state of the North Atlantic Ocean using ocean colour images. ☆

Florent.Garnier[a,*], Pierre.Brasseur[b], Jean-Michel.Brankart[b], Yeray.Santana-Falcon[b], Emmanuel.Cosme[b]

[a]*LEGOS, University of Toulouse, CNRS, IRD, CNES, UPS, (Toulouse), France*
[b]*Univ. Grenoble Alpes, CNRS, IRD, Grenoble INP, IGE, (Grenoble), France.*

**Abstract**

In this paper, we investigate the potential of using a probabilistic modelling approach in the prospect of ocean colour data assimilation. The main objective of the study is to assess the benefits of using error covariances based on an explicit simulation of model uncertainties. The relevance of this approach is evaluated by considering 3D observational updates of the ensemble (one update at one time step) performed every 5 days (over one year) using the statistics of a North Atlantic coupled NEMO/PISCES stochastic ensemble simulation involving 60 members, as previously described in Garnier et al. (2016).

In this experiment, SeaWIFS ocean colour data are used to update the ensemble with a low rank ensemble Kalman Filter analysis scheme. The non-Gaussian behaviour of the model variables is taken into account using anamorphic transformations. Comparisons between the updated ensemble and the MERIS satellite observations shows that the integration of high resolution SeaWIFS data significantly improves the representation and the ensemble statistics of chlorophyll concentrations. We also show that these improvements consistently cascade in the water column chlorophyll distributions and on non-observed variables closely linked with the primary production.

In addition, we present first results illustrating the potential of our approach for biogeochemical forecasts. The objective is to examine the model response to data assimilation in the perspective of future operational applications. For this purpose, we perform a 60 member simulation initiated from updated biogeochemical states. This forecast simulation shows that ocean colour data assimilation would be skillful considering integration cycles of the order of a day. Finally, the intend of this article is to point out the feasibility of operational biogeochemical data assimilation in the near future.

*Keywords:*
*PISCES, ocean colour data assimilation, probabilistic approach, forecast*

---

☆This document is a collaborative effort from all authors
*Corresponding author
  *Email address:* `florent.garnier@legos.mip-obs.fr` (Florent.Garnier )



## 1. Introduction

Since the 1980's, observations from a sequence of ocean colour satellite missions (e.g. Eppley et al.,
1985; Platt and Sathyendranath, 1988; Antoine et al., 1996; Behrenfeld and Falkowski, 1997) are available
to characterize the timing, spatial distribution and amplitude of ocean primary production and seasonal
blooms. Modelling approaches have been developed in parallel, providing additional information to comple-
ment remote sensing data on the vertical (into the whole euphotic zone), the horizontal (between satellite
swaths or to compensate cloud coverage) and temporal (between revisit observation times) dimensions (e.g.,
Doney et al., 1996; Aumont and Bopp, 2006; Gruber et al., 2006; Ilyina et al., 2013; Aumont et al., 2015).
Considering today's fundamental needs for estimating stocks of carbon and marine resources in a climate
change context, a synthesis between these different sources of information including their respective accuracy
is required to reduce uncertainties on the representation of the marine biogeochemistry (and in particular
the primary production).

The generation of synthetic products describing the biogeochemical ocean state is one of the challenging
goals of the Copernicus Marine Environment Monitoring Service (CMEMS). One major issue is to develop a
capacity to routinely assimilate ocean colour data into coupled physical-biogeochemical models, and thereof
deliver real-time and reanalysis products of the global ocean biogeochemistry. Following the seminal mod-
elling approach proposed by Garnier et al. (2016) to develop a stochastic version of the PISCES model, we
examine in this paper the usefulness of ensemble biogeochemical simulations to make one step towards this
synthesis goal.

So far, the data assimilated in biogeochemical models have varied from phytoplankton light absorption
(Shulman et al., 2013), satellite diffusive attenuation coefficient (Ciavatta et al., 2014), ocean-colour plankton
functional types (Skákala et al., 2018; Ciavatta et al., 2018) or colored dissolved organic carbon (Gregg and
Rousseaux, 2016). However, most of the studies focused on the assimilation of chlorophyll $a$ concentrations
derived from ocean colour data because other essential variables of the marine biogeochemistry are nearly
unobserved. The assimilation techniques vary from optimal interpolation methods (Gregg, 2008), variational
methods (Losa et al., 2003) or Kalman Filter methods (Ciavatta et al., 2011b, 2016; Fontana et al., 2009;
Simon and Bertino, 2009b; Simon et al., 2015).

Today, the performance of ocean colour data assimilation remains limited by the ability of biogeochem-
ical models (BGCMs) to realistically represent the observed variability of biogeochemical properties. This
limitation results, in part, from the many sources of uncertainties associated to the approximate description
of the physical background, the reduced complexity of biogeochemical model, the unresolved scales or the
empirical parameterizations of biogeochemical processes. In fact, the description of the model uncertainties
(which are often referred to as "model errors") is a critical challenge of the data assimilation problem (Lahoz
et al., 2010). In this context, ensemble methods (e.g., Evensen, 1994, 2003) provide a statistical means to
describe uncertainties associated with a complex model system by simulating the evolution of the probability
density function (pdf) of the system. The introduction of random processes into the model equations can
be used to simulate the spread of the ensemble members introduced in a data assimilation scheme. In the
numerical weather prediction context, an increasing number of operational systems rely on probabilistic
ensemble simulations (e.g., Palmer, 2012; Berner et al., 2011; Houtekamer and Zhang, 2016), while ensemble
simulations are still quite unusual in oceanography.

For oceanic biogeochemical applications, Woods and Onken (1982) and Wolf and Woods (1988) ad-
dressed the question of biogeochemical modelling using an ensemble of Lagrangian particles to represent
phytoplankton populations. More recently, several studies (e.g., Dowd, 2011; Béal et al., 2010; Weir et al.,
2013) have used probabilistic configurations of coupled biogeochemical models (CPBMs) based on the intro-
duction of stochastic parameterizations that simulate the effects of unresolved or poorly-resolve processes
(see Leutbecher et al., 2017, for a review). With this approach, the effect of hypothesized uncertainties are
directly implemented into the model equations by introducing random numbers to simulate uncertainties.

In this context, the objective of the present study is to assess how the ensemble simulations, as produced by
Garnier et al. (2016), can be used in a data assimilation perspective. Compared to other studies, the origi-
nality of this approach is the use of ensemble statistics originating from the parameterization of modelling
uncertainties to compute ocean colour assimilation updates. Finally, the overarching goal is to test whether





the integration of ocean colour data into a probabilistic coupled simulation can improve the representation
of primary production, as a first step to build a complete ensemble ocean data assimilation system.

To achieve this goal, we first use the probabilistic North Atlantic 1/4° configuration of the NEMO-
PISCES coupled model (hereinafter NATL025- PISCES) fully described in Garnier et al. (2016) to generate
a 60-members prior ensemble simulation. Then, using high resolution SeaWiFS ocean colour data, we
compute a sequence of 5-day ensemble updates to investigate if a low rank ensemble Kalman filter analysis
scheme is able to reduce the model uncertainty. In particular we will verify if the analysis improves the
statistical consistency of the ensemble. I order to investigate the predictability of the system, some of the
biogeochemical updates are thereafter used as initial conditions to produce ensemble forecasts. The objective
is to assess whether the information brought by the observations is preserved with time during the forecast,
which is a necessary condition prior to data assimilation.

The paper is structured as follows: section 2 introduces the model configuration and the prior ensemble
simulation. The section 3 presents the ensemble analysis method and a sequence of three-dimensional
ensemble updates. The section 4 describes the potential of ocean colour observations in the perspective of
biogeochemical forecasting. Finally, a summary including conclusions and future perspectives are proposed
in Section 5.

## 2. Model configuration and prior ensemble simulation

### 2.1. The coupled physical-biogeochemical NEMO/PISCES Model configuration

All simulations have been performed with the primitive model equation NEMO (Nucleus for European
Modelling of the ocean, Madec et al. (2012)) platform in its version 3.4 to realistically represent the ocean
dynamics and its interactions with the biogeochemical components. The main part of the NEMO system
is the 3D hydrostatic OPA (Ocean PArallelised, Madec et al. (1998)) module which computes the oceanic
circulation. For the purpose of this study, OPA is coupled with the sea ice LIM-v2 model (Louvain-La-Neuve
sea Ice Model, Fichefet and Morales Maqueda, 1997) and the biogeochemical model PISCES-v2 (Aumont
et al., 2015).

The coupled NEMO/PISCES model configuration is based on the NATL025 configuration, a sub-element
of 1/4° global ORCA025 configuration developed within the framework of the DRAKKAR project (Barnier
et al., 2006) to ensure a realistic representation of the ocean dynamic. Since the physical representation
of the ocean is crucial for simulating primary production, an eddy-permitting 1/4° horizontal resolution
is used. This configuration includes 46 vertical levels and covers the North Atlantic basin from 20°S to
80°N and from 98°W to 23°E. In the context of biogeochemistry, the suitability of this eddy-permitting
framework has been demonstrated in previous studies using the LOBSTER model (e.g. Doron et al., 2011,
2013; Fontana et al., 2013).

The physical component is coupled with the biogeochemical PISCES-v2 model Aumont et al. (2015).
The PISCES architecture consists on 24 biogeochemical variables from 4 major compartments including 2
classes of phytoplankton (diatoms and nanophytoplankton), 2 classes of zooplankton (mesozooplankton and
microzooplankton), 5 limiting nutrients (iron, silicate, phosphate, ammonium, and nitrate) and a detritus
compartment composed of dissolved and particulate carbon. This model has been widely used in previous
environmental studies such as Rodgers et al. (2008); Brasseur et al. (2009); Steinacher et al. (2010); Tagliabue
et al. (2014). PISCES is computed to be able to simulate the large scale biogeochemical processes of the first
trophic level of marine ecosystems (Bopp et al., 2015) and is considered as a relevant solution to simulate
primary production at the North Atlantic basin scale.

To guarantee a realistic representation, the coupling between the physical and the biogeochemical models
is performed on-line. The coupling frequency is 40 minutes. Note that the biogeochemistry has no feedbacks
on the physics. The tracer advection is computed by the physical model using a MUSCL scheme (Monotone
Upstream Scheme for Conservative laws, (LeVeque, 2002). Tracer vertical diffusivity fluxes are calculated
with the Turbulent Kinetic Energy (TKE, Blanke and Delecluse (1993)) closure scheme. The horizontal
diffusivity is expressed with a laplacian operator.



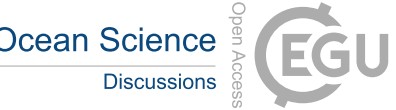

All the physical parameters used in the coupling are issued from a NATL025 simulation previously vali-
dated in Candille et al. (2015). The biogeochemical part is initialized in January 2002 from the MERCATOR-
OCEAN 1/4° coupled NEMO/PISCES model BIOMER simulation. Between January 2002 and December
2004, an additional 3 years spin-up is performed to ensure a consistent biogeochemical initial state, prior to
perform ensemble simulations used in this study.

### 2.2. The prior model ensemble chlorophyll estimation

In order to characterize model uncertainties, we introduce random processes ($\mathbf{w}_{t_k}$) into the model for-
mulations. The evolution to the biogeochemical state $\mathbf{x}_{t_k}$ by the model $\mathcal{M}$ is then described by equation 1.
The biogeochemical state temporal dynamic depends on the random perturbation sequence of $\mathbf{w}_{t_k}$ generated
here from first order auto-regressive processes (AR1) including a spatial correlation of 5 grid points and a
decorrelation time of 30 days. The $\mathbf{w}_{t_k}$ random perturbations are then introduced into the model formulation
through the stochastic parameterizations proposed in Garnier et al. (2016) to simulate the uncertainties on
biogeochemical parameters and the uncertainties induced by unresolved scales in the presence of non-linear
processes. Similar approaches (e.g., Ciavatta et al., 2016) use random perturbations over model forcing
parameters in order to define the model error. With these methodologies, perturbations are only applied to
the initial state. By contrast, using the present formulation, the co-variances statistics are time dependent
and defined from the model dynamics which improves the prior model error definition.

$$\mathbf{x}_{t_k} = \mathcal{M}_{t_{k-1}, t_k} \left( \mathbf{x}_{t_{k-1}}, \mathbf{w}_{t_k} \right) \quad ; \quad \mathbf{x}(0) = \mathbf{x}_{t_0} \tag{1}$$

Using this probabilistic approach, we perform a 1-year 60-member ensemble simulation during 2005
which is briefly shown here. For this purpose, figure 1 shows the ensemble quartiles of surface chlorophyll
along 20°W longitudinal sections together with MERIS observations. Quartiles are calculated at each grid
point from the prior ensemble probability distributions. It therefore does not display a model solution but
a composite between the 60-member of the ensemble.

Because of strong biogeochemical dynamics, the dispersion is higher during the spring bloom period.
We also observe that the level of dispersion is very space and time dependent. It tends to be maximum
during the spring bloom, in the northern part of the domain and in the equatorial current, where chlrophyll
concentration are generally high. Note that around the 20° latitude region, the very high dispersion level is
due to the presence of the Mauritania's upwelling. In some particular geographical areas, the biogeochemical
activity can be very weak. Chlorophyll concentrations are close to zero and the ensemble is nearly non-
dispersive. In spite of this large spread, the median is nearly always very close to the deterministic standard
simulation (*detIni*) described in Garnier et al. (2016), proving that PISCES main large scale dynamics is well
preserved. The main difference with respect to a deterministic simulation is that most of the observations
(about 70% over the whole domain) are well included within the ensemble envelope, which means that the
stochastic parameterizations generate a level of uncertainty consistent with the variability of the observations.
The ensemble quantiles also provides a rough vision of the shape of the probability distribution. The range
of values of the 50% higher chlorophyll concentrations (above the median) is always larger than the 50%
range of smaller concentrations (below the median) which underlines the non-gaussianity of the probability
distribution. This point will be taken into account in the assimilation process using an anamorphosis
transformation (cf section 3.1)

It is necessary to specify that envelopes presented in figure 1 depict the statistical information imple-
mented as the model error in the analysis scheme presented in the next section. Because it is constructed
from model solutions, it directly includes a spatial and temporal representation of its variability. Since
these results indicate that the ensemble display a sufficient level of uncertainty to capture the information
content of the observations, the model error definition should also be consistent with the variability of the
observations. In addition, such an evolutive characterization of model uncertainties present a robust tech-
nique which avoid to increase model co-variances when the dispersion tends to be very small (in particularly
outside seasonal bloom periods).



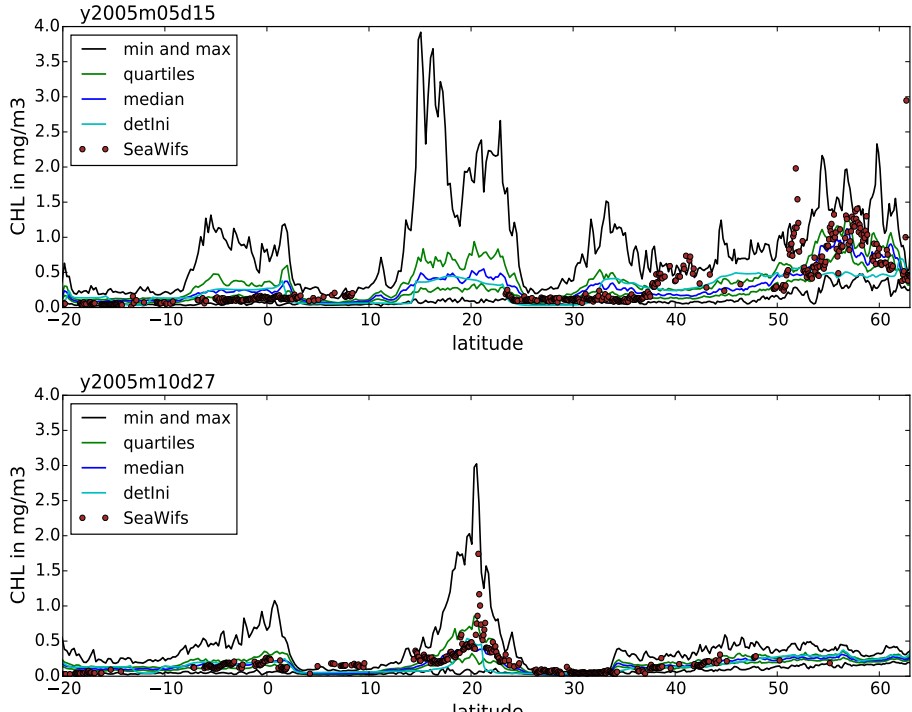

Figure 1: Surface chlorophyll quartiles along a $20°$W longitunal section for the $15^{th}$ of May and the $27^{th}$ of October 2005. Maxima and minima are represented in black lines. The $1^{st}$ et $3^{rd}$ quartiles are represented in green and the median is in dark blue. Red dots indicate SeaWIFS observations. the deterministic *detIni* standard PISCES simulation, performed without stochastic parameterizations (Garnier et al., 2016), is represented in cyan.

## 3. A sequence of 3D ensemble update

In this part, the prior probability distributions are used to update the model biogeochemical state. We present a sequence of analysis meaning that updated states are not propagated by the model. The objective is to assess the benefits of using prior covariances based on a explicit simulation of model uncertainties.

### 3.1. Methodology/observations

#### 3.1.1. The ensemble analysis method

The analysis presented in this article is based on the sequential Ensemble Transform Kalman Filter (ETKF, Bishop et al., 2001). The analysis scheme involves a transformation matrix to calculate the covariances in the space of anomalies $\delta x_i^f$ (size $s \times N$) which are calculated from the ensemble mean state $\overline{x^f}$ (equation 2) of the model forecasts $\{x_i^f : i = 1, 2..., N\}$ here $N = 60$) following the equation 3.

$$\overline{x^f} = \frac{1}{N} \sum_{i=1}^{N} x_i^f \qquad (2)$$





$$\delta\mathbf{x}_i^f = \mathbf{x}_i^f - \overline{\mathbf{x}}^f \tag{3}$$

These anomalies characterize the matrix $\mathbf{S}^f$ of size $i$ which defines the prior covariance matrix $\mathbf{P}^f$ following equation 4. Note that with this formulation, error distributions are subject to the gaussian hypothesis.

$$\mathbf{P}^f = \mathbf{S}^f\mathbf{S}^{f^T} \quad with \quad \mathbf{S}_{(i)}^f = (1/\sqrt{N-1})\delta\mathbf{x}_i^f \tag{4}$$

The Kalman gain matrix ($\mathbf{K}$) is calculated in the reduced space of anomalies from the product of the anomaly matrix $\mathbf{S}^a$. The change of space is made by the transformation matrix $\mathbf{T}$ expressed by equation 5, where $\mathbf{H}$ is the observation operator to pass through the observation space. In the observation space, the anomalies are therefore expressed by $\mathbf{Y}^f = \mathbf{H}\mathbf{S}^f$. $\mathbf{R}$ is the diagonal observation error covariance matrix.

$$\mathbf{P}^a = (\mathbf{I} - \mathbf{K}\mathbf{H})\mathbf{P}^f = \mathbf{S}^a\mathbf{S}^{a^T} = \mathbf{S}^f\mathbf{T}\mathbf{T}^T\mathbf{S}^{f^T} = \mathbf{S}^f\left(\mathbf{I} + \mathbf{Y}^f\mathbf{R}^{-1}\mathbf{Y}^{f^T}\right)^{-1}\mathbf{S}^{f^T} \tag{5}$$

From an eigendecomposition, the matrix $\mathbf{\Gamma} = \mathbf{Y}^f\mathbf{R}^{-1}\mathbf{Y}^{f^T}$ can be expressed by equation 6 from the unitary matrix $\mathbf{U}$ and the diagonal matrix $\mathbf{\Lambda}$ consisting of the eigenvectors and the eigenvalues of $\mathbf{\Gamma}$

$$\mathbf{\Gamma} = \mathbf{U}\mathbf{\Lambda}\mathbf{U}^T \tag{6}$$

We now define the transformation matrix $\mathbf{T}$ which calculates the updated anomalies and the ensemble mean.

$$\delta\mathbf{x}_i^a = \mathbf{S}^f\mathbf{T} \quad avec \quad \mathbf{T} = \sqrt{N-1}\mathbf{U}(\mathbf{I} + \mathbf{\Lambda})^{-1/2}\mathbf{\Lambda}^{1/2}\mathbf{U}^T \tag{7}$$

$$\overline{\mathbf{x}^a} = \overline{\mathbf{x}^f} + \mathbf{S}^f\mathbf{U}(\mathbf{I} + \mathbf{\Lambda})^{-1}\mathbf{Y}^{f^T}\mathbf{R}^{-1}\left(\mathbf{y}_0 - \overline{\mathbf{H}\mathbf{x}^f}\right) \tag{8}$$

The step of analysis as it will be done here is then only an updated of the ensemble members:

$$\mathbf{x}_i^a = \overline{\mathbf{x}^a} + \delta\mathbf{x}_i^a \tag{9}$$

In this scheme, the gaussianity of the prior probability distributions remains a strong hypothesis. To consider the non-gaussian shape of chlorophyll distributions, a solution is to apply a log-normal transformation. However, using such a space in time invariant transformation is generally insufficient (Ciavatta et al., 2011a). In this study, the non-gaussianity of the probability distribution is taken into account using anamorphosis transformation (Béal et al., 2010) depending on the shape of the probability distribution. The principle is to characterize a bijective transformation to redesign the percentiles of the distribution in such a way to produce a gaussian probability distribution. The ETKF analysis scheme is thereafter calculated in this anamorphosis space. In the context of biogeochemical data assimilation, this method has already been successfully employed in several studies such as Simon and Bertino (2009a); Doron et al. (2011, 2013); Fontana et al. (2013).

Although a stochastic approach is a relevant solution to characterize model uncertainties, the sampling of the covariance matrix from 60 members can be a source of unrealistic large distance spatial correlations. In an area like the North Atlantic where horizontal dimensions are much larger than the correlation lengths, this phenomenon must not be neglected. For this purpose, the spatial extent of corrections is reduced by introducing a domain localization matrix (Testut et al., 2003). Model covariances are multiplied by a correlation function $\gamma(r) = \exp(-r^2/l^2)$ which exponentially attenuates spatial correlations as a function of the distance $r$. The cut-off radius $l\_c$ defines the distance after which the covariances are imposed to zero. The values $l = 4$ and $l_c = 16$ grid points, inherited from the work of Candille et al. (2015), are used.





### 3.1.2. Ocean colour observations

In this study, chlorophyll concentrations assimilated in the coupled model are derived from the SeaW-
iFS sensor launched on-board the SeaStar-Orbview2 satellite. More specifically, we use a high resolution
(1/24°) version of the SeaWiFS data set-up in the ESA (European Spatial Agency) project OC-CCI (Ocean
Colour Climate Change Initiative). These data are accessible through the MyOcean-Copernicus web site
*(http://marine.copernicus.eu)*. In addition, 1/12° MERIS-Envisat ocean colour observations are used to
perform statistical assessments.

In the analysis scheme described in section 3.1.1, we suppose 1) a gaussian distribution of observa-
tion errors and 2) that the observation are spatially uncorrelated. Consequently, $\mathbf{R}$ is a diagonal matrix
characterized by a 30% observation error. In order to take into account the non-gaussianity of chloro-
phyll distributions, the observations $\mathbf{y}^0$ and their associated standard deviations are also evaluated into the
anamorphosis space defined by the ensemble simulation. Note that the anamorphosis transforms marginal
distributions into gaussian distributions but it does not guarantee the gaussianity of joint distributions
(between variables).

Using high resolution observations, a particular attention is given to the validity of the uncorrelated
observation hypothesis. Indeed, in spite of its requirement in the assimilation scheme, errors between near
observations points can not be entirely uncorrelated. Without considering these correlations, the model
information is overwhelmed by the redundancy of the observations. To take into account correlation of
errors while maintaining a diagonal matrix, we inflate the diagonal terms of the observation covariances
matrix proportionally to an estimated distance of the spatial correlations. A constant correlation distance
of 2.3°, determined experimentally, is set. The equation 10 presents these parameters where $\sigma$ is the standard
deviation and the $\alpha$ constant refers to the square root of the number of data observations contained within
a 2.3 ° radius circle around the grid point.

$$\mathbf{R} = \alpha(\sigma^2\mathbf{I}) \quad with \quad \sigma = 0.3 \times CHL \quad and \quad \alpha = 2.3 \times 24 \tag{10}$$

### 3.2. The Updated ensemble

Using the method described previously, the prior 60-member model simulation (cf. section 2.2) is cor-
rected every 5 days during the entire year 2005. Hence, the results correspond to a 3D sequence of ocean
colour data analysis (73 biogeochemical updated states).

### 3.2.1. Impact on the chlorophyll mean state

Figure 2 presents surface chlorophyll maps of the prior ensemble mean, the updated ensemble mean and
the assimilated SeaWiFS observations for the $31^{th}$ of March, the $15^{th}$ of May and the 27 $^{th}$ of October.
These maps demonstrate that the analysis mostly impacts the mean state during the spring bloom (15 $^{th}$ of
May). Compared to the prior solution, the analysis increases chlorophyll concentrations over the subtropical
gyre and decreases it in the Gulf Stream and the Northern regions. Consequently, the strong north to south
gradient existing in the prior representation is smoothed, which results into an enlargement of the subtropical
oligotrophic conditions. In general, the updates reduce deviations with observations but a more detailed
comparison with observations is necesseray. It will be performed in next sections.

On average, the analysis do not induce abrupt variations of the large scale chlorophyll surface distribu-
tions and the solution always complies with the large scale biogeochemical dynamics defined by the prior
model. In the context of data assimilation, this point is essential to maintain information brought by the
analysis during propagation steps and does not prevent the model representation to include small scale
variability. Thus, these results especially: 1) ensure the relevancy of the analysis method, and 2) confirm
the consistency of the prior spatial surface chlorophyll correlations.

To go further, figure 3 presents 30°W chlorophyll vertical sections during the spring bloom (15 $^{th}$ of May)
and the autumn season ($27^{th}$ of October). Note that vertical extrapolation of only surface observations is





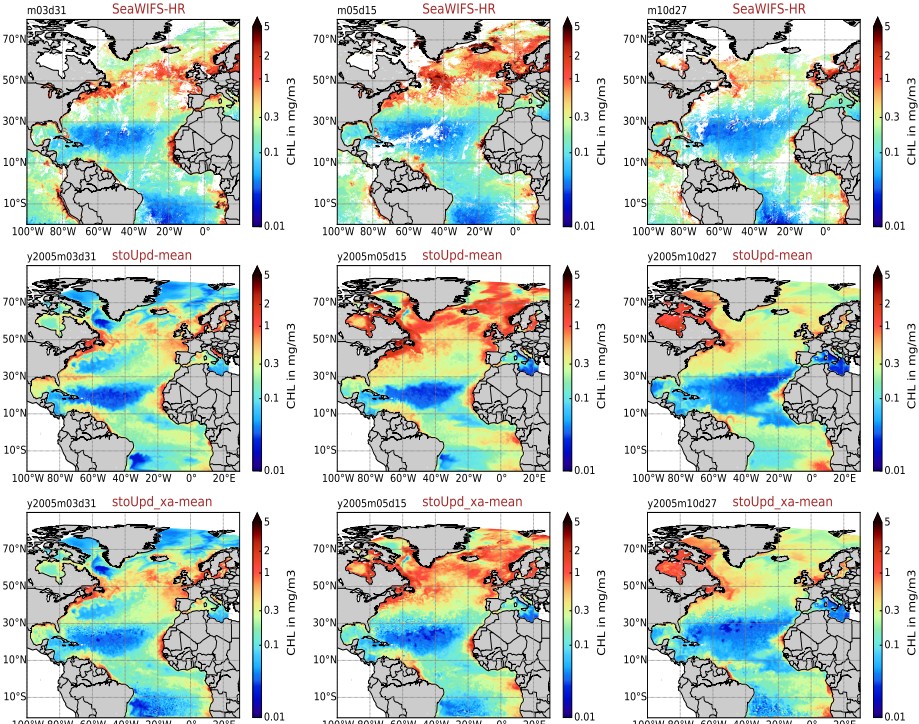

Figure 2: Surface chlorophyll maps of the prior mean state (*StoUpd-mean*, $2^{nd}$ line), the updated mean state (*StoUpd_xa-mean*, $3^{rd}$ line) and the SeaWIFS observations ($1^{st}$ line) used to constrain the model . Results are shown for the 31 March ($1^{st}$ column), the 15 May ($2^{nd}$ column) and the 27 October ($3^{rd}$ column) of 2005.

a major challenge in data assimilation. The first point to observe is that the analysis has an impact in the whole euphotic zone. In the equatorial zone and in the region located around 40°N, the analysis reduces the chlorophyll concentration in the first 50 meters. On the contrary, after 40°N, it is reduced. A remarkable feature is that surface chlorophyll decreasings are associated with an enhancement the DCM (Deep Chlorophyll Maxima). These results highlight an important correlation between surface chlorophyll concentration and vertical chlorophyll distribution. Below the euphotic layer, the analysis does not produce adverse effects.

In spite of these impacts, the important point is that the analysis does not affect, on average, the vertical characteristics such as the depth of propagation and the localization of subsurface maxima. which shows that the method doesn't seem to induce unrealistic correlations on vertical fluxes (for example vertical organic matter fluxes). Such as for the surface, the analysis preserve main elements of the PISCES model vertical dynamics, to which the validity was previously verified in Garnier et al. (2016). The use of vertical correlations based on our stochastic parameterizations is therefore able to describe relevant extrapolations of ocean colour data onto the vertical.




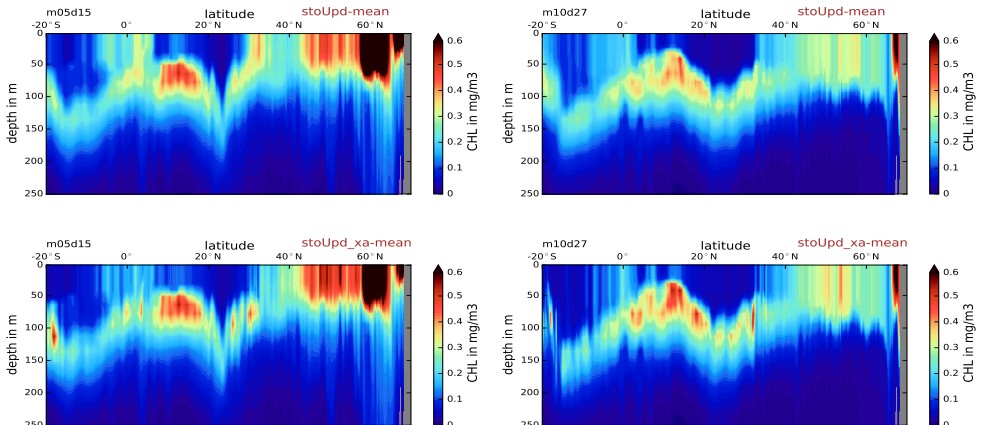

Figure 3: Chlorophyll vertical sections at 30°W for the prior mean state ($StoUpd\text{-}mean$, $1^{st}$ line) and the updated mean state ($StoUpd\_xa\text{-}mean$, $2^{nd}$ ligne) for the 15 May ($1^{st}$ column) and the 27 October ($2^{nd}$ column) of 2005.

### 3.2.2. Uncertainty of the analysis

Using a probabilistic approach, a measure of the uncertainty must take into account the various shapes of probability distributions. To investigate the dispersion without gaussian hypothesis, the figure 4 presents 60-member surface chlorophyll time series of the prior and the updated ensembles at 2 grid points. The STAT_B grid point is located in the equatorial region and the STAT_E is representative of the regime of the Northern part of the domain.

This figure shows that the analysis reduces the dispersion while preserving a significant level of uncertainty. This point is fundamental as it will characterize the prior error model in an assimilation scheme. Rationally, the analysis mainly impacts high chlorophyll concentration patterns for which prior uncertainties are higher. Therefore, the system acts mainly during the high biogeochemical activities of the spring bloom period. Observations remain most of the time close to the median solution and are nearly always captured by updated ensemble spread. Note that updated members are also included within the prior ensemble envelope. It means that chlrophyll is only consistently corrected from a prior distributions in good agreement with observations. In this sense, a good definition of the stochastic parameterization appears is essential.

As regards to the vertical dispersion, figure 5 presents vertical chlorophyll profiles of the 60-member of the prior and the analyzed ensembles at one grid point ($STAT\_C$) located in the Gulf Stream region. As for the surface, the analysis reduces the dispersion while keeping a significant degree of uncertainty throughout the euphotic layer. It is relevant to observe that the 2 different biogeochemical regimes (in spring and automn) are kept during the analysis. The dispersion seems to be higher for the depths of subsurface maxima and during the spring bloom, when model uncertainties are higher. At a global scale, the level of uncertainty is only slightly reduced which is related to the strong level of biogeochemical uncertainties rather than a failure of the system. In this case, this is quite positive since a small dispersion would have been detrimental to the assimilation process. Finally, assuming relevant prior ensemble definition, i.e relevant stochastic parameterization, surface chlorophyll corrections should be consistently reflected into vertical chlorophyll distributions.

Using the proposed probabilistic method in assimilation processes, the introduction of ocean colour data





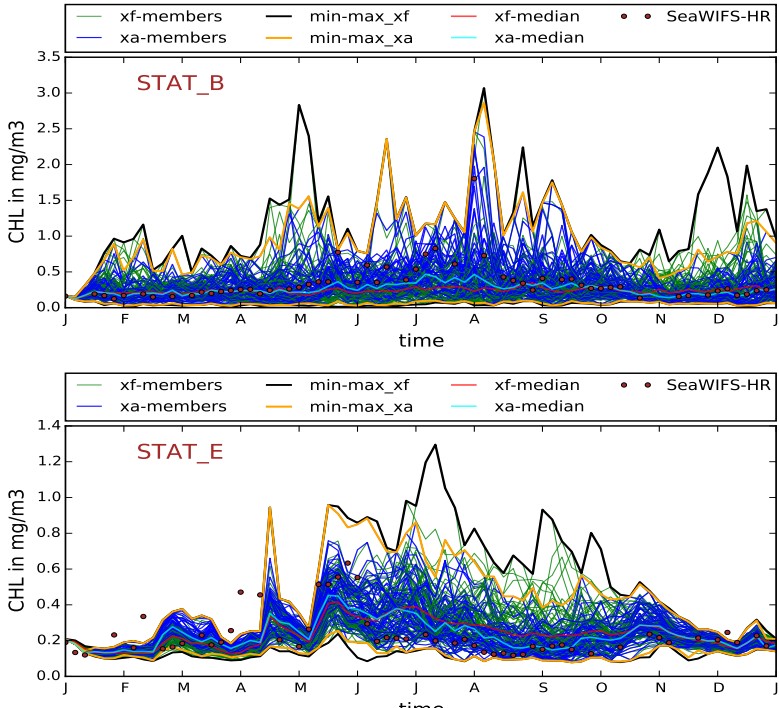

Figure 4: Surface chlorophyll time series of the 60-member of the prior ensemble members (*xf-members*, green) and the xa updated ensemble members (*xa-members*, blue) in the substantive *STAT_B* (located in the equatorial zone) and *STAT_E* (located in the Northern part of the domain) grid points. The high resolution SeaWIFS observations used for the analysis are indicated in red circles. The statistical envelope is defined by the quartiles of the ensemble. For the prior ensemble, minima and maxima (*min-max_xf*) are in black and the median (*xf-median*)is in red. For the updated ensemble, minima and maxima (*min-max_xa*) are in yellow and the median (*xa-median*)is in cyan.

should be able to consistently correct chlorophyll fields. Since stochastic parameterizations characterize an prior ensemble with relevant statistics, co-variances computed from the model dynamics avoid to drive the model toward incoherent biogeochemical states. Obviously, we also need to ensure that this method can sufficiently constrain the biogeochemical system to be considered for data assimilation.



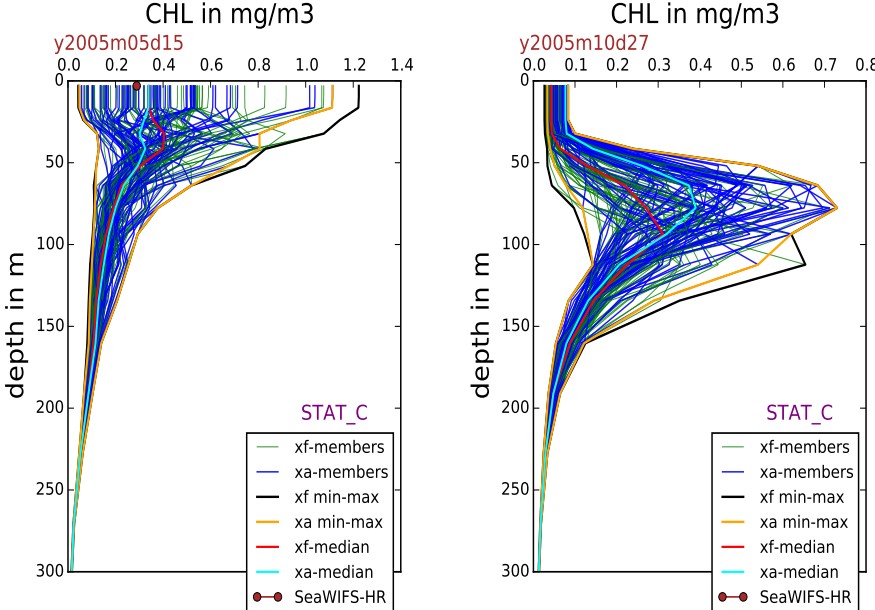

Figure 5: Chlorophyll vertical profiles of the 60-member of the prior xf ensemble members (green) and the xa updated ensemble members (blue) in the substantive $STAT\_C$ grid point for the 15 May (left) and the 31 October (right). The surface SeaWIFS observation is indicated as a red circle. The statistical envelope is defined by the quartiles of the ensemble.

### 3.3. Probabilistic assessment

In this section, impacts of the analysis on the ensemble distribution are investigated from a statistical comparison with MERIS ocean colour observations. Two statistical properties are used: the reliability, which evaluates the level of consistency between the ensemble probability distributions and the statistical distribution of observations and the resolution which measures the distance between the cumulative probability distribution of the ensemble and the observations expressed in terms of probability distributions. Further information is available in Candille and Talagrand (2005) and Candille et al. (2015). Here, these properties are deduced from the rank of observations (rank histograms, Anderson (1996)) and from the Continuous Rank Probability Score (CRPS, Brown (1974)).

### 3.3.1. Ranks of MERIS observations

The reliability is evaluated from the repartition of MERIS observations within the ensemble range of chlorophyll concentrations (the rank of observations), displayed in the form of histograms. The flatness indicates the level of reliability: a flat histogram depicts a perfect reliability. As decribed in Saetra et al. (2004), ranks are calculated taking into account a 30% observation error by increasing the ensemble dispersion. The reliability investigated here is a spatial reliability in the sense that the distribution of observations is deduced from their spatial variability at a given date.

In figure 6, the shape of the prior ensemble rank histogram is compared to that of the updated ensemble during the spring bloom. All available observation over the domain are considered. The rank histogram of





the prior ensemble display an underdispersive behaviour (about 80% of the observations are included) with a distinctive ∪ shape reflecting the difficulties of the ensemble to encompass the observations. The analysis clearly reduces this underdispersion. The updated ensemble capture approximatively 10% more observation and its histogram is strongly flattened, which indicates that the introduction of SeaWiFS information improves the reliability.

In addition, the spatial distribution of the ranks is much more heterogeneous for the updated ensemble. Depending on the geographical zone, this highlights local improvements on the reliability and a better mid-range representation of chlorophyll concentrations. The updated ensemble ranks also present higher small scale variability that seems to be in better agreement with the variety of biogeochemical behaviours existing at a basin scale. Nevertheless, as for the prior simulation, many observations (found around the Gulf Stream and the subtropical gyre regions) still remain outside the envelope of the ensemble.

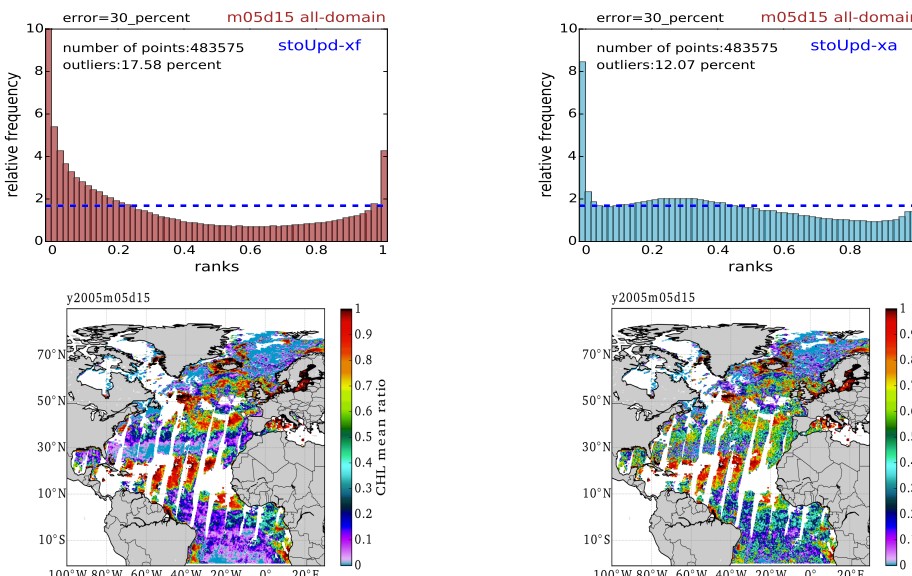

Figure 6: Surface chlorophyll ranks of the MERIS ocean colour observations calculated for the 15 $^{th}$ may 2005 using all available data on the domain. A 30% observation error is considered. Top panels present the ranks from histograms while the bottom panels present spatial maps of the ranks. The first column displays the prior ensemble ranks (*stoUpd-xf*) and the second column shows ranks of the updated ensemble (*stoUpd-xa*). The frequency of occurence is given in percent (over the total number of observations). The blue dotted lines in rank histogram figures refer to the rank repartition of a perfectly reliable ensemble.

### 3.3.2. The continuous ranked probabilistic score

The continuous ranked probability score (CRPS) is a generalization to continuous events of the Brier score (Brier, 1950). As expressed by equation 11, it corresponds to the distance between the ensemble cumulative probability density function $F_p(x)$ of a $x$ variable (here the chlorophyll) and the probability distribution $F_o(x)$ of its observation $y_o$ (here the ocean colour data).

$$CRPS = E\left[\int_{\mathbb{R}} (F_p(x) - F_o(x))^2 dx\right] \tag{11}$$





where $E[.]$ is the mean operator over all verifications (the ocean colour data available in the North Atlantic basin at a given date). The probability distribution $F_o(x)$ is therefore a Heaviside function defined by equation 12. Note that we don't consider observation errors for this diagnosis.

$$H(x) = \left\{ \begin{array}{lll} 0 & si & x < y_o \\ 1 & si & x \geq y_o \end{array} \right. \tag{12}$$

In order to differentiate the statistical properties it is possible to decompose the CRPS as the sum of 2 terms: CRPS=RELI+RESO. The CRPS decomposition is fully documented in Hersbach (2000). RELI refers to the ensemble reliability and RESO refers to the potential resolution, that is the resolution of the ensemble in the case of a perfect reliability. The closer to 0 is the value the better is the score.

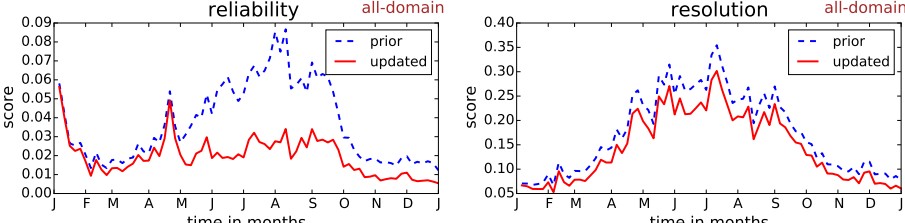

Figure 7: Surface chlorophyll time series of the reliability and the resolution of the prior ensemble (blue) and for the updated ensemble (red), calculated from the Continuous Ranked Probability Score (CRPS) decomposition. At each date, all available MERIS observations are used for the calculations.

Using all MERIS observations available every 5 days during 2005, we calculate the RELI and the RESO terms of the CRPS decomposition. Results are given in time series in figure 7.

As already observed with rank histograms, it shows that the analysis systematically improves the global reliability of the ensemble. This result is more moderate during the first months of the year because the dispersion coming from the stochastic parameterizations has not yet been entirely generated. In average the reliability is improved of about 70% in summer and of about 30% in winter which confirms the results observed with the rank histogram. The higher biogeochemical activity (and so the ensemble dispersion) during the spring and the summer period probably mainly explain this difference. Indeed, deviations between the model and the observations are higher during this period and the analysis has also higher impacts.

Concerning the resolution, improvements due to the analysis are smaller and remain relatively constant. The analysis improves the resolution for about 15-20% This is the result of a slight narrowing of the probability distribution extent which mainly causes a decreasing of extreme events probability of occurrence.

Finally, the analysis never induces negative impacts on the ensemble statistics, which underline the consistency of our method.

### 3.4. correction on non-observed variables

So far, we have focused on chlorophyll concentrations. Indeed, the stochastic parameterizations used to generate the ensemble aimed to simulate some source of uncertainties centered on primary production. One of the major issue in data assimilation is,nevertheless, the correction of non-observed variables. This means that we need to verify that the integration of chlorophyll data could cascades onto these non-observed variables To address this issue, the figure 8 presents 60-member surface time series for 3 variables closely linked with primary production, the nitrates, the phosphates and the microzooplankton. The aim is only to ensure that the co-variances inherited from the stochastic parameterizations produce updated biogeochemical states in accordance with the prior information.





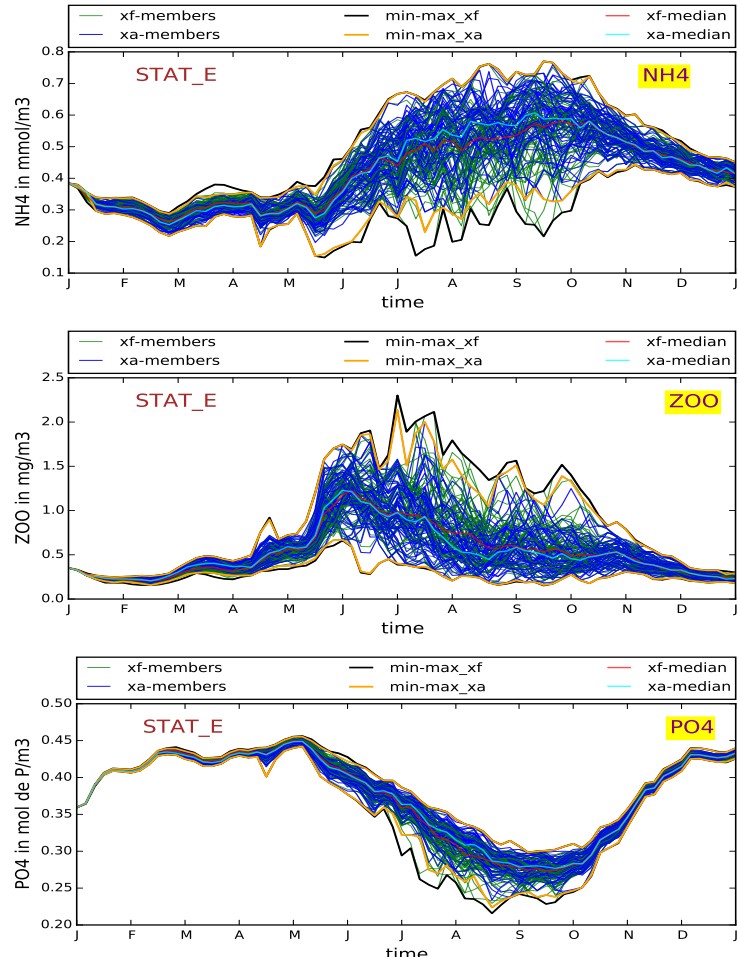

Figure 8: Surface time series of the 60-member of the prior ensemble (xf-members) and the updated ensemble (xa-members) in the substantive $STAT\_E$ grid point representative of the biogeochemical regime of the Northern part of the domain. Fom the top to the bottom results are presented for the ammonium (NH4), the microzooplankton (ZOO) and the phosphate (PO4). The colour code is reference in the figure and kept identical to the one of figure 4.

This figure demonstrates that the stochastic parameterizations also induce dispersion onto the non-observed variables (of the prior simulation). In return, the method of analysis is able to affect their statistical distributions and to reduce the ensemble dispersion. As for the chlorophyll, all updated members are kept inside the prior envelope, which means that prior characteristics are conserved after the analysis and highlight the importance of an appropriate definition of prior model statistics. Finally it is relevant to

underline that the correlations established from the stochastic parameterizations are able to correct some non-observed variables closely related to primary production. This point is crucial in order to constrain the biogeochemical state in the North Atlantic basin scale using only chlorophyll satellite data.




Nonetheless, a comparison with observation, as it has been done for the chlorophyll, would be neces-
sary to assess the validity of the prior statistics and the following corrections. As long as the stochastic
parameterizations would be defined in such a way they can generate consistent dispersion for the concerned
non-observed variables, the corrections should be relevant. The question which still held open: would the
analysis performs with only one observed variable be sufficient to correctly constrain a biogeochemical state
characterized by many variables ? In order to give first elements of response, we perform forecast expriments
presented in the next section.

## 4. Potential of ocean colour observations in the perspective of biogeochemical forecasting

The approach presented here consists in producing ensemble forecasts initialized from updated biogeo-
chemical states. The objective is to investigate the model response to the integration of ocean colour
information in order to define data assimilation approaches adapted to the context of marine biogeochem-
istry. Furthermore, another aspect addressed here is the ability of the system to accomplish relevant short
and/or long term biogeochemical forecasting.

### 4.1. Impact of the analysis on the prior ensemble distributions

Updates of the ensemble were previously computed every 5 days. In order to make consistent forecast
experiments, a finer temporal resolution is essential. Indeed, improvements due to the analysis could be
rapidly deteriorated because of the strong level of uncertainty assumed for the PISCES model. We therefore
compute daily outputs during a 1-month period of the spring bloom (5 April 2005 to 5 May), known
to display strong discrepancies between model and observations. For this experiment, two 60-member
ensemble simulations have been carried out. The first simulation is initialized by the multivariate updated
biogeochemical state presented in the previous section, while the second is set to be identical to the prior
simulation (cf. section 2.2), except that the outputs are daily.

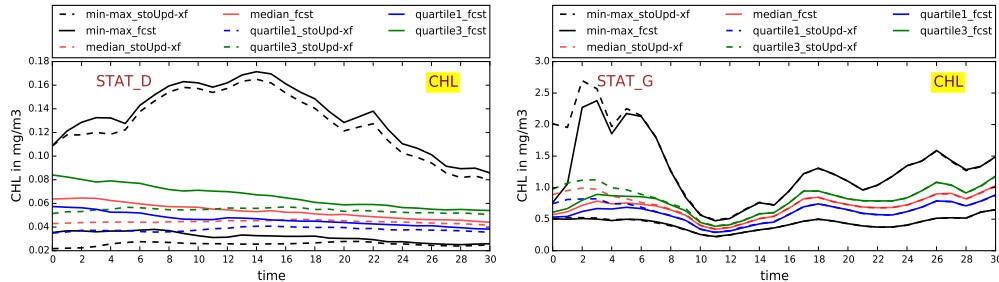

Figure 9: Surface chlorophyll time series of the ensemble prior quartiles (in dot lines) and the forecast quartiles (full
lines) at the $STAT\_D$ grid point located in the equatorial region and the $STAT\_G$ located in the Northern part of
the domain. The x-axis refers to the time (in days) between the 05/04/2005 until the 05/05/2005.

Figure 9 presents 1-month daily time-series of the ensemble quartiles for the prior simulation (dotted
lines) and the forecast simulation (full line) in two grid points highlighting typical behaviours. Depending
on the geographical area (more precisely on the biogeochemical dynamics), the ocean colour information is
generally sustained during several days by the ensemble (e.g at $STAT\_G$). A maximum of about 1 month
(e.g at $STAT\_D$) was detected. This seems to reveal a short term predictability of the system, probably
due to the strong biogeochemical model uncertainties. The tendency to rapidly converge towards the prior
solution illustrates the stability of the PISCES model. Indeed, PISCES is strongly constrained by the value





of many parameters and the model mostly tends to converge toward the solution for which it was initially
calibrated. Note that forecast members tend toward their free corresponding member because the exact
same random numbers are used in the free simulation and the forecast.

### 4.2. Comparison with observations

Taking into account our statistical approach, it is essential to assess the consistency between the forecast
ensemble and ocean colour observations. Two issues have to be investigated: (1) for how long an update is
able to reduce the prior level of uncertainty, and (2) how does it affect the statistical characteristics of the
prior probability distributions. To investigate the first issue, figure 10 shows a 1-month time serie of the
root mean square error (RMSE) for the forecast and the prior simulation. The RMSE is calculated every
day using the mean of the two ensembles and the mean of all the available MERIS observations. Note that
observations used for the calculation are spatially varying with time.

As expected, the figure shows that the integration of ocean colour data reduces the system level of
uncertainty as compared with the prior simulation. It also confirms the short-term predictability of the
system. According these results, the decreasing of the RMSE due to the analysis vanishes after 5 days but
a significant reduction of the uncertainty is only kept for 2 days. In a context of data assimilation, this
indicates that it is necessary to integrate ocean colour data with high frequency to consistently constrain
the model.

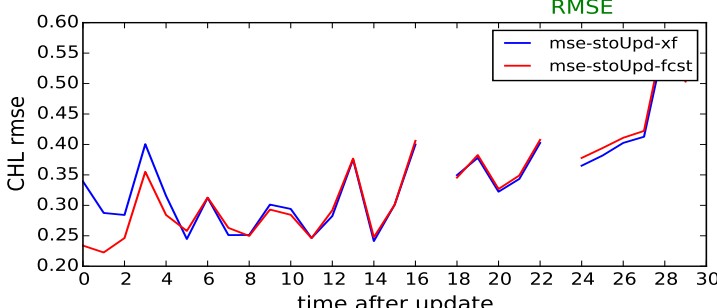

Figure 10: Time series of the root mean square error (RMSE) using the MERIS ocean colour observations. The
blue curve present the prior ensemble and the red curve present the forecast. Each day, all observations available in
the domain are used for the calculation. The x-axis refers to the time (in days) between the 05/04/2005 until the
05/05/2005.

In section 3.4, we have shown that the analysis improves the statistical coherence between the ensemble
and satellite observations. The second question to address is whether this feature can be preserved during
the forecast. As before, we compare the ensemble simulations with observations using rank histograms of
440 MERIS observations which are presented in figure 11. Note that calculations of the ranks are identical to
those presented in figure 6.

After 1 day of forecast (6 April 2005), it is remarkable to see that the shape of the histogram remains
nearly unchanged. Statistical improvements due to the analysis are then preserved during this period.
Beyond this time, histograms become comparable again with those of the prior ensemble. After 5 days of
445 prediction, the rank histograms of the forecast and the prior ensemble are almost identical. The gain of
reliability coming from the analysis has been lost. Because of a high level of uncertainty, the biogeochemical
model very rapidly generates some dispersion which make ensemble forecasts no more reliable than the free
run after 1 day.




In summary, the analysis improves the predictions of the mean representation of surface chlorophyll for a few days but the ensemble forecast remains reliable only after one day of forecasting. Using our framework, this means that data assimilation could only be efficient performing daily cycles. Next step would be to set-up a complete data assimilation system that integrates ocean colour information to daily update the forecast. In this manner we could evaluate whether statistical improvements seen in the present study are
preserved in time or not, even though nowadays satellite products lack of a complete spatial coverage.

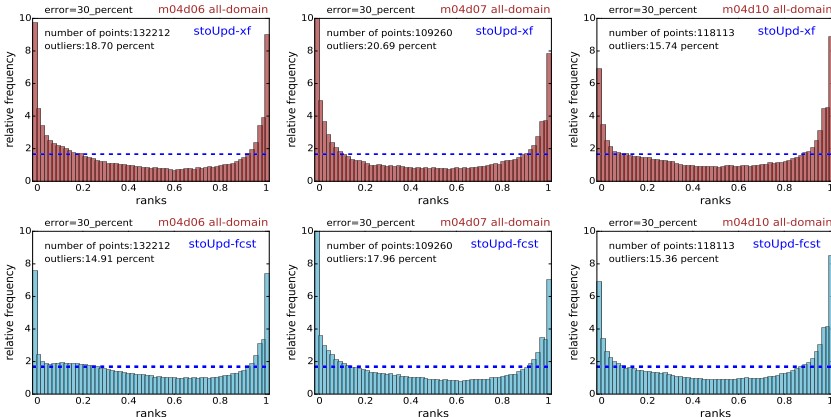

Figure 11: Surface chlorophyll rank histograms of the MERIS ocean colour data calculated for the whole domain with a 30% observation error. Results are shown for the 06/04/2005 (1$^{st}$ column), 07/04/2005 (2$^{nd}$ column) and the 10/04/2005 (3$^{rd}$ column). Top panels present the histograms of the prior ensemble (*stoUpd-xf*) and the bottom panels those of the forecast (*stoUpd-fct*). The frequency of occurrence is given in percent (over the total number of observations) The blue line refers to the rank of a flat histogram.

## 5. Summary and perspectives

The ensemble approach developed in this paper is a first step towards ocean colour data assimilation for ocean biogeochemistry, with a focus on the representation of primary production. Using the probabilistic version of the PISCES model assessed in Garnier et al. (2016), a time-dependent integration of ocean
colour images based on stochastic parameterizations has been documented using a 60-member ensemble simulation. The main objective is to demonstrate the potential of this approach to assimilate ocean colour satellite observations into realistic coupled physical/biogeochemical model configurations.

Our results demonstrate that the proposed model error parameterization is appropriate for biogeochemical data assimilation context. In particular, the method is shown to significantly improve the statistical
consistency (indicated by reliability and resolution) of the ensemble. Furthermore, the uncertainty of the updated biogeochemical states is reduced while keeping the dynamical characteristics defined by the prior ensemble simulation. These outcomes indicate that the methodology is promising for future operational applications. Nonetheless, some issues need to be further explored.

According to our forecast experiments, the implementation of ocean colour data assimilation is skillful over short time cycles of about one or two days. Beyond this time, improvements due to the integration of observations into the biogeochemical model become negligible with the currently used PISCES formulation. The dynamics of chlorophyll so appears to be strongly constrained by the prescribed biogeochemical PISCES model parameters and forecasts tend to quickly converge towards the prior ensemble trajectories. An
important feature highlighted is the importance of the initial definition of stochastic parameterizations, that



define the model covariances. The efficiency of the analysis strongly depends both on a relevant choice of the biogeochemical sources of uncertainties, and of an appropriate level of uncertainties. In order to improve the assimilation of biogeochemical data, it is therefore suggested to refine stochastic paramaterizations, relatively to the integrated observations. Further experiments should allow to investigate whether using parameter estimation techniques could sustanaibly constrain the primary production.

In practice, the proper duration of the ocean colour data assimilation window in an operational system should in the order of a day, while it is in the order of the week for altimetric data assimilation in a mesoscale circulation model. A question that arises from the present study is therefore the controllability of biogeochemical systems. First, it is important to remind that the surface chlorophyll is the only observed variable over 24 that make up the PISCES model. Despite the use of a multivariate scheme with a relevant definition of correlations, fundamental processes such as the vertical nutrient fluxes are not be sufficiently monitored to constrain the phytoplankton growth in the whole euphotic zone. An issue that need to be investigated is therefore the introduction of vertical profiles in addition to surface information. The integration of *in situ* physical variables data into ocean circulation models have already been investigated in earlier studies (e.g., Griffa et al., 2006). Similarly, other biogeochemical variables should be incorporated to coupled systems. For this purpose, data provided by BIO-ARGO floats (Johnson et al., 2009; Johnson and Claustre, 2016) are essential to consolidate the observing system. Meanwhile, this kind of requirements for data assimilation is an important feature to promote the development of observation networks. The main obstacle for using verticale profiles is the need for sufficient space and time coverage to generate large scale impacts From an algorithmic point of view, the localization scheme used in our analysis will be an issue to assimilate vertical profiles as it could only impact a small radius around the observation points whereas an effective large scale constrain would be required to extract the information of various scales contained in these observations. However, the correction of large scale nutrient fluxes and surface chlorophyll concentrations would presumably improve the model representation of primary production.

On the other hand, uncertainties coming from the physical part of the system are not taken into account in the present configuration despite the critical role of the physics on biogeochemical fluxes. In our system, physics is considered as deterministic and certain. However, the vertical upcoming of nutrients in the euphotic zone depends on the vertical oceanic currents and vertical mixing induced by various physical sources. Thus, it is necessary to correct the physics in order to constrain biogeochemical vertical fluxes. In this sense, uncertainties coming from both the physical and the biogeochemical parts of the system have to be considered. Note that ensemble twin experiments are being develop to address this question (e.g., Yu et al., 2018), and future attempts with the present methodology would help in that task.

**Acknowledgements**: This work has received funding from the European Community's Seventh Framework Pro- gramme FP7/2007-2013 under grant agreements 283367 (MyOcean2) and the CMEMS 36-GLO-HR-ASSIM funded by EC, with additional support from CNES. It is also a contribution to the CNRS/INSU/LEFE program. The calculations were performed 610 using HPC resources from GENCI-IDRIS (grant 2015-011279). The authors are grateful to the two anonymous reviewers for their relevant and constructive comments.

.

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
