# Peer review of "An ensemble probabilisitic approach to reconstruct the biogeochemical state of the North Atlantic Ocean using ocean colour images. $\stackrel{\diamond}{\approx}$"

_Ocean Science, 2018_

## Referee Comment (RC1) · Anonymous Referee #1 · 13 Feb 2019

The paper is extremely interesting, and clearly very beneficial for the future assimilation of surface chlorophyll in BGC models, including future stochastic operational systems. The paper heavily relies on Garnier et al, 2016, and builds on the ensemble experiment to use it for data assimilation. In this aspect, it clearly deserves publishing. However, a number of issues need to be resolved first. There issues do not require to re-do the scientific work, but rather to reformulate some parts of the paper more carefully. In my opinion, some of the re-writing is really necessary for the paper to be accepted.

General comments

[Figure]

The paper is confusing at the first lecture because it lacks a few capital (basic) pieces of information in the first sections. One needs to read it entirely to find that information, or re-read it to fully understand. Some of the missing information is very superficially mentioned, without being definitively clear about it,and one is left more confused than if nothing was said at all.

The most striking example is the fact that the authors mention on line 171 (page 6) that "In this part, the prior probability distributions are used to update the model biogeochemical state. We present a sequence of analysis meaning that updated states are not propagated by the model." I noticed that, but did not understand exactly what it meant. Do you produce $x^a$, maybe use it to compute statistics etc, but then actually discard it and continue the model run using $x^f$ ? Then page 15 (9 pages later), you introduce another run where you compare 2 ensembles during a 1-month run. One is the same as previously, and the other probably is one where you actually use $x^a$ to restart the model ? Even that is not clear. The text lines 75-80, 170-171 and 400-405 should be made much clearer in that respect.

I understand that ensemble simulations at this scale, including a Bio model, require a lot of computing power, and I don't have problems with you discarding $x^a$ for subsequent model runs (if that's what you do). For the 1-year run, it could be that you computed the analysis from the [Garnier et al 2016] outputs, and did not actually re-run the ensemble of models. That is OK, but it is a highly unusual way of working, and if it's the case, then it should really be explained clearly.

Another such confusion is what you actually put in the state vector x introduced around line 130. Is it the whole biogeochemical model (all 24 3D variables) as you seem to imply with "biogeochemical state"? Is it NCHL and DCHL ? Is it the sum of NCHL and DCH ? Suddenly at line 404, we see "multivariate". Also you don't explain anywhere (unless I missed it) how you link the chloro observation (y) with the state vector (x). Is it H that performs the sum of NCHL and DCHL ? Intuitively, if y>Hx, how do you split the increment onto the variables inside x ? In the same proportions of what they already

contained ? All this needs to be clarified.

Finally, the paper sometimes presents affirmations, of the kind "we see A, so this implies B", which, in my opinion, are not valid. They will be listed below among the specific comments.

Specific comments and typos

Line 75-80: this is already related to my first general comment, to lines 130, and should really be clarified much more explicitly, so that the reader unequivocally knows what you are doing, what you are using to restart the model every 5 days, and what you are analyzing in the statistics. For example, to me, the "update" (line 81) means the "increment", not the "analysis" itself (i.e.: $x^a = x^f + update$ ), so I'm confused when you say "some of the BGC updates are used as initial conditions". The word "some" is confusing me even more.

Line 113: PISCES is computed to be able to ...: please formulate elsehow.

Line 117: the coupling frequency is 40 minutes. "frequency" is unfortunate but common, and we understand what you mean, so that's ok. In Garnier et al 2016, you coupled at every time step. Do you now try to save cpu time ? Because you did re-simulate the whole ensemble after all? What's the impact of this 40-minutes coupling compared with the coupling in Garnier et al 2016 ?

Line 130: please define more precisely what you put in x . Define also H, here or when you introduce it around line 184

Line 148: [my comment probably outside this article's scope] Would the conclusion about the ensemble dispersion in upwelling region be different, if you also perturbed physics?

Line 153: "the main difference with respect to a deterministic simulation" –> this is not a "difference", because in a deterministic simulation we are not speaking about whether observations are in an envelope, or not. In a deterministic run, we need to use different

(poorer) metrics. So, I understand your point, but this needs to be rephrased.

Line 155: this is an example of invalid "A–>B". The fact that the observations are all in the envelope (or 70% of them) does NOT imply that your ensemble has a consistent level of uncertainty. I can make an ensemble, instead of CHL, I use an envelope which is {0 Chl+100000}. Now ALL observations will be inside my envelope. If you really want to say that the level of uncertainty is consistent, you could check how many of the observations fall into the different percentiles. An example (for physics, not BGC) is given in Vandenbulcke and Barth, 2015 ( 10.1016/j.ocemod.2015.07.010 ), when we can suppose the variable is Gaussian, than we know how many observations SHOULD fall into the different percentiles. Obviously in your case, a priori you don't know the real PDF shape

Line 163-165: Since...content of the observations ==> the model error ...observations. Again, this is not a consequence. Furthermore, in line 165, please avoid to use "error" for the model but "variability" for the observations.

Line 167: what you say is logic, and I agree. But even if you choose another method (not a multiplicative one), I suspect that the effect of DA on the model state would be very small when CHL is close to zero, because I expect both x and y would be close to zero. This is not a critic of your method!

Section 3 title: add "s" to "update"

Line 171: not clear, see general comments

Equation 7: translate "avec"

Equation 9: if xˆa contains the sum of NCHL and DCHL, explain how you upgrade the model variables once you have xˆa. If xˆa contains NCHL and DCHL themselves, explain how the analysis equation takes this into account (I suppose that H is doing the summation over the 2). This is the same comment as the before-last general comment

Line 195: replace "in" with "and" Line 196: add "an" at the end of the line

Line 203, remove the line (except the last 2 words), it's obvious

Line 216: remove "MyOcean", the reader will not find that anymore, just Copernicus / CMEMS

Line 233 : I agree with the method to inflate the diagonal to account for missing non-diagonal elements in R. Maybe you can cite e.g. Brankart et al (2009) or Cosme et al (2013) who propose other methods (tri-diagonal etc), but it's not mandatory. You could also link what you do to the representativity error (observations represent smaller scales than the model). In any case, maybe replace "refers" with "is proportional to" ? Else, in equation 10, wouldn't you multiply by 3.14?

Line 248-249: Replace by: In general, the updates reduce deviations with observations. A more detailed comparison with observations will be performed in next sections.

Line 250: The ensemble average does not show abrupt variations, but maybe individual members do? Small scales are usually averaged-out of ensemble means. You could show some members to prove your point, or not (your choice).

Lines 250-255: You seem to imply that "Line 250-254" imply "254-255". Again, I do not agree. Again, I could give you a stupid counter-example: use the most stupid and defect method in the world, but choose R=diag(10000000). Small scales and large scales *will* be preserved in the analysis. It does not "ensure" and "confirm" what you say. But I agree that it is a good sign and gives confidence in the method. I don't criticize the method, but I suggest to rephrase more carefully

Line 256: Remove "To go further", it doesn't help with anything.

Line 260: after 40°N, it is reduced –> you probably mean increased, not reduced ?

Line 262. Where in the plot is the DCM "enhanced" ? I don't see that (maybe I'm not looking where I should. What do you mean by 'enhanced' anyway? Stronger absolute values of chlorophyll ? Deeper DCM ? In general, the enhancement (or lack of it) of your DCM is "just" what your correlations indicates (the P matrix, or more precisely

the columns of S, the members of your ensemble). Whether the enhancement is right or not, who knows (unless you showed us some observed vertical profiles, that you probably don't have). So we cannot know if the actual correlation (from the ensemble) between surface and deeper layers in the euphotic zone are actually right (by looking at the DCM enhancement). It sure is reassuring that you obtain zero correlation between surface and depths>euphotic.

Line 263. correlation between surface "concentration" and vertical "distribution", please rephrase better

Line 266: use capital 'W' on 'which'

Line 267. Here it would have really helped me if I knew what was inside the vector x, and how the experiment is working. If it's only Chl, or alternatively, the model does NOT restart from $x^a$, but just from $x^f$, then it is normal that the model does not generate vertical fluxes of organic matter in between assimilation cycles. If necessary, you can rephrase this

Line 270. Ok, I agree with this

Line 275: remove *the" before "figure 4"

Line 279-280: It seems to me that whether you preserve or not a "significant" (please define what is significant, or use another word) level of uncertainty in $P^a$, is a consequence of both $P^f$ (your ensemble) and R. Furthermore, even if $P^a$ was small (compared to $P^f$), it could very well be that the model increases it in the same amount until the next assimilation cycle, isn't it ?

Line 283 : if you think it's helpfull, you could give a number here. We know that for the prior envelope, 70% of observations are included. You don't give the equivalent for the posterior envelope, you just say "nearly always".

Line 284: add "the" before "updated ensemble spread"

[Figure]

Line 285: It means that ... . Same comment, it doesn't necessarily means that. You have a good and convincing situation, but A does NOT imply B.

Line 286 : obvious (but you should keep the line if you think it helps the reader)

Figure 4 (and other similar figures): many lines, difficult to see. The xf-members are hidden by the xa-members? But I don't have any suggestion to do better than that. At least the min,max,median are visible and convincing

Figure 4 legend: add a space after ")" (2 times)

Line 293-294: add "model" to "uncertainty" in line 293, at the replace "biogeochemical uncertainties" with something like "uncertainty on the observations".

Line 295: Maybe it is positive, maybe not. Maybe the model will greatly increase P ? Maybe not, in this case you're right. The reader cannot fix his ideas because [see general comments]

Line 297: obvious (but you should keep the line if you think it helps the reader)

Line 301, last word: replace "an" with "a" Lines 300-304: I would move this to the conclusions, or rephrase

figures 5. In May it seems the DCM is decreased, in October increased. Is this what you meant with 'enhanced' ?

Figure 6 (lower panels) and Lines 330-332 : this is not obvious from the figure. Can you explain ?

Line 354: Suddenly you say that the first months of the year are not reliable ? Isn't there a spin-up as in Garnier et al 2016 ? Isn't it possible this rather (or also) has to do with seasonality (reliability goes down again from the end of the summer) ? If the first months aren't reliable, what about all the previous things you said in the article, what about the plots you showed for 31/March, etc ? Maybe you are saying that the reliability in the first 4 months should actually look like the one in the last 3 months ?

That except the seasonality effect, the remainder of the (lack of) difference between prior and posterior reliability is due to (lack of) spin up ?

Section 3.4 Title : Use capital 'C'

Line 368: put a space before "nevertheless" Line 369: remove "s" from "cascades"

Section 3.4. Here it would have been good to know for sure, from the start of the article, if x contains chlorophyll or all model variables

Line 379: yes if there are correlations between variables, then observing one can "correct" them all. If you put all variables into x, then I bet you will have correlations even between not-so-closely-related variables, exactly as we have unphysical long-range correlations in space (which we remove with localization techniques). So yes, we can "correct", but until we have observations, I'm not convinced yet that we are improving anything. In the EnKF, the analysis step is still a linear process (although the propagation of P by the model is not).

Line 383 on the next page. Ok now I agree.

Line 386, "should be relevant", yes indeed they should. I'm really curious now if the EnKF is correcting the non-observed variables (through the multivariate covariance matrix and state vector) or if the model is (by propagating corrections of chlorophyll to other variables).

Line 388: add "some" before "first" and "the" before "forecast"

Line 400: if you're using a 5-day window, the analysis could be rapidly deteriorated, but what do you mean with the strong level of uncertainty for PISCES ? Do you mean that you added large stochastic terms in the equations, or that PISCES (even deterministic) has a tendency to rapidly generate errors (e.g. drifting always toward the same "wrong" solution as you write later in the article) ?

Section 4, in general: the term "forecast" is may be confusing, it usually means a

model run without assimilation (because it's in the future), as opposed to hindcasts and analysis

Figure 10 and discussion lines 425-430: do you assimilate only at time=0 in this experiment ? Obviously at some point, you restarted the model from x̂ᵃ, and then let it run for one month ?

Line 474: yes I agree with that ! also, "sustanaibly"

Line 476: ... should BE OF the order of a day, while it is OF the order of ...

Line 479: "over 24" –> "out of 24"

Line 479 : multivariate ...

Line 480: are not "be" sufficiently –> remove "be"

Line 498: "to correct": I disagree with the word choice. I would rather say it is necessary to take the uncertainty of the physics into account. Whether you can correct it or not depends on observations, etc. But if you represented the uncertainty in your ensemble, at least you would account for it.

I remember a talk from K. Fennel where she explained that perturbing BGC without perturbing also the physics, did not work for her (maybe she says something similar in the paper that you cite, Yu et al 2018)

Title : I personally find the title long and maybe a little redundant. For example, "ensemble" and "probabilistic" are redundant. "to reconstruct" is also sort of useless. However I recommend to the author to keep this title if he so wishes.

––––––––––––––––––––––––––––

---

## Referee Comment (RC2) · Anonymous Referee #2 · 27 Feb 2019

The manuscript examines the assimilation of satellite chlorophyll data into an ocean-biogeochemical model with stochastic parameterization. In particular the study uses the outputs of an ensemble run with stochastic parameterization to conduct single-step assimilations over one year. Thus, the assimilation is not cycled in this case, but each assimilation is applied to the ensemble state from the previous ensemble simulation. In addition, the ensemble forecast over one month initialized from the analysis ensemble on April 5, 2005 is used to assess the influence of the assimilation on the model forecast skill. The authors conclude that the assimilation improves the chlorophyll field

estimate and the ensemble. Further the method updates some nutrients and subsurface chlorophyll in a stable way. For the forecast case, a stable positive influence of the assimilation is found for one day.

When one reads the manuscript and is aware of the current state of research in the assimilation of satellite chlorophyll data into a biogeochemical model, it is difficult to determine what is actually new in this work. There are already several studies that assimilate satellite chlorophyll data and that actually do perturb parameters of the biogeochemical model (Hu et al. 2012, Doron et al. 2013, Ciavatta et al. 2016, Jones et al. 2016, Pradhan et al. 2019). However, the exact way how the perturbations are applied are different from that used here. To this end the manuscript shows that the stochastic perturbation approach of Garnier et al. (2016) leads to ensemble covariances that are usable for data assimilation (at least for a single step). On the other hand, the data assimilation setups of other studies include cycled data assimilation and are hence far more advanced than the single-step data assimilation methodology applied here. From my experience, one cannot deduce a stable successful data assimilation process from a single analysis step. Thus, the data assimilation experiments of the manuscript can only indicate that the real cycled assimilation might be successful (because the single analysis step looks promising). With regard to the forecasting experiment, the finding that the assimilation leads to better forecast over one day is far shorter than what other studies find. This might indicate that the perturbation method applied here is in fact not usable for operational biogeochemical data assimilation, in contrast to the claim at the end of the abstract. Over all, I cannot recommend the manuscript for publication in the present form. Below I provide some recommendations, which might enhance the study sufficiently to be relevant enough for publication.

Recommendations:

I find it essential that the authors perform cycled data assimilation. A single analysis step (or many single analysis steps) do not allow to conclude that a cycled assimilation process (and this is required to get significant improvements of the model state over

Interactive
comment

time) will be successful.

With the focus of the authors to 'assess the benefits of using error covariances based on an explicit simulation of model uncertainties' and 'The relevance of this approach is evaluated' (lines 8-9) it will be relevant to compare the proposed stochastic perturbation scheme with standard schemes, e.g. that applied by Doron et al (2011), of which some authors are the same as in the manuscript. A 'benefit' or 'relevance' can only be determined in comparison to other methods, which is completely missing the manuscript.

The authors also need to perform a verification of their results with independent data. Actually, the statement '... SeaWIFS ocean colour data are used to update the ensemble...' (line 13) and that MERIS satellite data is used for comparison (line15), suggests an independence which is actually not present. According to the manuscript, OC-CCI satellite data obtained from the Copernicus web site is used for the assimilation. The OC-CCI data is actually a combination of data from several satellites (SeaWiFS, MERIS and MODIS in 2005). Thus MERIS data is actually part of the OC-CCI data set and hence the authors verify their assimilation influence with regard to a part of the assimilated data. (This fact is actually not too obvious from the Copernicus web site, but the original OC-CCI data files (www.oceancolour.org) contain the information which satellites contribute to the data)

With regard to the description of the methodology it is irritating that the authors explain the full ETKF method in Sec. 3.1.1 but avoid any details on the anamorphosis transformation. Actually, the ETKF is a widely-used standard method has already been explained in many papers. In contrast, the anamorphosis methodology is much less common. To this end it would be much more relevant to explain the details of the anamorphosis method instead of showing the equations of the ETKF. This similarly holds for the decomposition of the CRPS score in Sec. 3.3.2. Here, the manuscript provides the definition of the CRPS, but only mentions that the CRPS can be decomposed into the terms ensemble reliability and potential resolution. Then, the authors

then show only the two decomposition terms. However, without the definition of the two decomposition terms and their proper explanation, it is impossible for a reader to actually interpret Fig. 7, which shows the effect of the assimilation on these terms.

In the experimental setup, I find the handling of the observation error very unusual. First, a observation error of 0.3 is used (line 227). This value is never motivated. It was the error target for SeaWiFS, but actually, the OC-CCI data include spatially varying error information. I'm wondering why this is not used in the experiment. Further, the authors describe that to account for possible error correlations in the observations, the observation error variances are inflated, but the observation error covariance matrix is kept diagonal. While I have seen such argumentation before, the authors miss to provide any reference for this choice. As such, the approach looks fully ad hoc here without theoretical foundation of the method. The inflation of the observation error variance is then performed with a factor of 55.2 (2.3 x 24), such that the assimilation uses an observation error variance of 4.968. I cannot remember any study assimilating chlorophyll data with such an extreme value, and I cannot follow the argumentation of the authors to perform this inflation (Ciavatta et al. (2016) describe the handling of the observational errors, based on OC-CCI product user guide, when averaging them onto a coarser resolution grid. However, this does not account for correlated errors) Actually, the extreme inflation of the observational error variance should lead to a very small influence of the assimilation. However, from the description of the results this is not evident. This might actually imply that the ensemble is too disperse, because only a large ensemble variance can counter a large observation error variance in the assimilation. Fig. 4 indicates that this might be the case. In particular at STAT_B (described cursorily as 'in the equatorial region' in line 283) we see minimum ensemble values very close to zero (seemingly below 0.1 mg/m^3) while at the same time values up to about 2.5 mg/m^3 at reached by other ensemble members. Given that this happens at the same location at the same time, the ensemble appears to represent so vastly different condition that it doesn't seem to have a reasonable skill here (which seems to contradict the histogram in Fig. 6, which shows on under-dispersive ensemble on average over the whole model domain). For STAT_E, also shown in Fig. 4, the ensemble spread is lower, but still the values seem to vary by a factor of 8 (between 0.1 and 0.8 mg/mˆ3), which is also large. The authors should clearly give the theoretical basis for the inflation and should more clearly describe how the observations are used (I have the impression that the used observation operator does interpolate from the 1/4deg model grid to the observation resolution of 1/24deg (OC-CCI is usually provided at 4 km resolution) without any super-obbing).

I further recommend that the authors perform a more thorough literature research. The manuscript does not leave the impression that the authors are really aware of the current status of the research on the assimilation of satellite chlorophyll data into ocean-biogeochemical models with ensemble data assimilation methods. E.g. Nerger and Gregg (2007, 2008) have already shown successful chlorophyll assimilation with an ensemble Kalman filter. Hu et al. (2012) demonstrate a successful assimilation with a rather highly-resolved regional model. Further, any reference to papers by Ford et al. (2012, 2017), current papers by Ciavatta et al. (2018) and Skakala et al (2018) are missing. Also the study by Pradhan et al (2019) should be relevant. It is also essential that the author relate their results in the subsections of Sec. 3 to already published findings. This is completely missing at the moment even more leaving the impression that the authors are not aware of the current status of this research. This even concerns studies like Doron et al. (2011, 2013) which are referred to in the methodological part, but their findings are ignored in Sec. 3. Relating the results to other studies does e.g. immediately show that the improvement of forecasts over just one day is extremely short (e.g. Pradhan et al. show that 5-day forecasts are still much better than the free running ensemble, but actually for a coarse global model. However, Ciavatta et al. use a much higher resolved model, but obtain longer forecast influence, and in some cases even assimilate only once per month.).

Finally, I like to recommend that the authors are more careful in their formulations. E.g. 'using ocean colour images' used in the title is incorrect. Used are neither 'images' nor

'ocean colour', but concentration maps of chlorophyll, which are derived from ocean color. Also 'we present first results illustrating the potential of our approach for biogeochemical forecasts.' (line 20) is misleading. Indeed, these are the 'first results' in which the chlorophyll assimilation is used with the perturbation method of Garnier et al. (2016). However, this ignores all other published studies, as the results described in the manuscript are not at all the first results in which chlorophyll is assimilated to improve the forecasts of biogeochemical model fields and of course not the first results pointing to the potential operational use of biogeochemical data assimilation. The authors leave a very odd impression of being 'first', while the assimilation methodology presented in the manuscript is in fact behind the state-of-the-art.

Given these significant shortcomings of the manuscript, I omit minor comments, like grammatical errors or unclear sentences. Anyway, I have the impression that the perturbation method used in the study can have potential, but this would need to be carefully shown.

References: Ciavatta et al. JGR Oceans 123 (2018) 834-854 Ford et al, Remote sensing of Environment 203(2017) 40-54 Ford et al., Ocean Sci. 8 (2012) 751-771 Hu et al., J. Mar Syst. 94 (2012) 145-156 Jones et al., Biogeosciences 13 (2016) 6441-6469 Nerger & Gregg, J. Mar. Syst. 68 (2007) 237-254 Nerger & Gregg, J. Mar. Syst. 73 (2008) 87-102 Pradhan et al., JGR Oceans 124 (2019) 470-490 Skakala et al., JGR Oceans 123 (2018) 5230-5247
* * *